# On Formal Feature Attribution and Its Approximation

## Abstract

Recent years have witnessed the widespread use of artificial intelligence (AI) algorithms and machine learning (ML) models. Despite their tremendous success, a number of vital problems like ML model brittleness, their fairness, and the lack of interpretability warrant the need for the active developments in explainable artificial intelligence (XAI) and formal ML model verification. The two major lines of work in XAI include *feature selection* methods, e.g. Anchors, and *feature attribution* techniques, e.g. LIME and SHAP. Despite their promise, most of the existing feature selection and attribution approaches are susceptible to a range of critical issues, including explanation unsoundness and *out-of-distribution* sampling. A recent formal approach to XAI (FXAI) although serving as an alternative to the above and free of these issues suffers from a few other limitations. For instance and besides the scalability limitation, the formal approach is unable to tackle the feature attribution problem. Additionally, a formal explanation despite being formally sound is typically quite large, which hampers its applicability in practical settings. Motivated by the above, this paper proposes a way to apply the apparatus of formal XAI to the case of feature attribution based on formal explanation enumeration. Formal feature attribution (FFA) is argued to be advantageous over the existing methods, both formal and non-formal. Given the practical complexity of the problem, the paper then proposes an efficient technique for approximating exact FFA. Finally, it offers experimental evidence of the effectiveness of the proposed approximate FFA in comparison to the existing feature attribution algorithms not only in terms of feature importance and but also in terms of their relative order.

## 1 Introduction

Thanks to the unprecedented fast growth and the tremendous success, Artificial Intelligence (AI) and Machine Learning (ML) have become a universally acclaimed standard in automated decision making causing a major disruption in computing and the use of technology in general [1, 29, 35, 47]. An ever growing range of practical applications of AI and ML, on the one hand, and a number of critical issues observed in modern AI systems (e.g. decision bias [3] and brittleness [64]), on the other hand, gave rise to the quickly advancing area of theory and practice of Explainable AI (XAI).

Numerous methods exist to explain decisions made by what is called black-box ML models [46, 48]. Here, *model-agnostic* approaches based on random sampling prevail [46], with the most popular being *feature selection* [56] and *feature attribution* [40, 56] approaches. Despite their promise, model-agnostic approaches are susceptible to a range of critical issues, like unsoundness of explanations [21, 24] and *out-of-distribution sampling* [34, 62], which exacerbates the problem of trust in AI.

An alternative to model-agnostic explainers is represented by the methods building on the success of formal reasoning applied to the logical representations of ML models [42, 61]. Aiming to address the limitations of model-agnostic approaches, formal XAI (FXAI) methods themselves suffer from a few downsides, including the lack of scalability and the requirement to build a complete logical

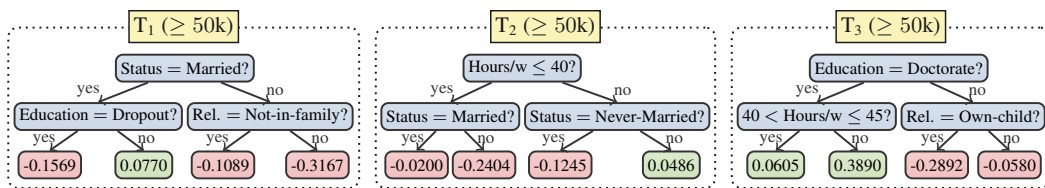

Figure 1: Example boosted tree model [12] trained on the well-known *adult* classification dataset.

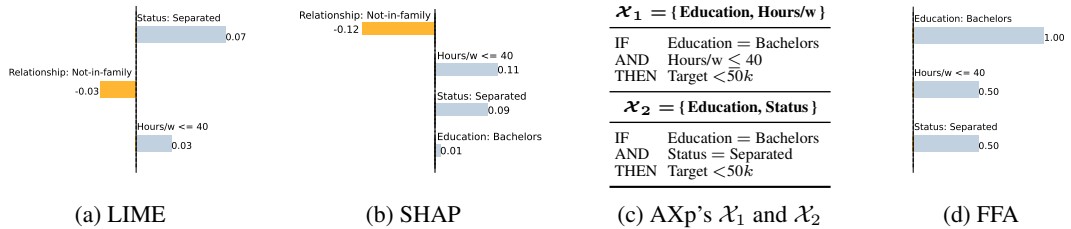

| (a) LIME | (b) SHAP | (c) AXp's $\mathcal{X}_1$ and $\mathcal{X}_2$ | (d) FFA |

Figure 2: Examples of feature attribution reported by LIME and SHAP, as well as both AXp's (no more AXp's exist) followed by FFA for the instance $\mathbf{v}$ shown in Example 1.

representation of the ML model. Formal explanations also tend to be larger than their model-agnostic counterparts because they do not reason about (unknown) data distribution [65]. Finally and most importantly, FXAI methods have not been applied so far to answer feature attribution questions.

Motivated by the above, we define a novel formal approach to feature attribution, which builds on the success of existing FXAI methods [42]. By exhaustively enumerating all formal explanations, we can give a crisp definition of *formal feature attribution* (FFA) as the proportion of explanations in which a given feature occurs. We argue that formal feature attribution is hard for the second level of the polynomial hierarchy. Although it can be challenging to compute exact FFA in practice, we show that existing anytime formal explanation enumeration methods can be applied to efficiently approximate FFA. Our experimental results demonstrate the effectiveness of the proposed approach in practice and its advantage over SHAP and LIME given publicly available tabular and image datasets, as well as on a real application of XAI in the domain of Software Engineering [45, 52].

## 2 Background

This section briefly overviews the status quo in XAI and background knowledge the paper builds on.

### 2.1 Classification Problems

Classification problems consider a set of classes $\mathcal{K} = \{1, 2, \ldots, k\}$[1], and a set of features $\mathcal{F} = \{1, \ldots, m\}$. The value of each feature $i \in \mathcal{F}$ is taken from a domain $\mathbb{D}_i$, which can be categorical or ordinal, i.e. integer, real-valued or Boolean. Therefore, the complete feature space is defined as $\mathbb{F} \triangleq \prod_{i=1}^{m} \mathbb{D}_i$. A concrete point in feature space is represented by $\mathbf{v} = (v_1, \ldots, v_m) \in \mathbb{F}$, where each component $v_i \in \mathbb{D}_i$ is a constant taken by feature $i \in \mathcal{F}$. An *instance* or *example* is denoted by a specific point $\mathbf{v} \in \mathbb{F}$ in feature space and its corresponding class $c \in \mathcal{K}$, i.e. a pair $(\mathbf{v}, c)$ represents an instance. Additionally, the notation $\mathbf{x} = (x_1, \ldots, x_m)$ denotes an arbitrary point in feature space, where each component $x_i$ is a variable taking values from its corresponding domain $\mathbb{D}_i$ and representing feature $i \in \mathcal{F}$. A classifier defines a non-constant classification function $\kappa : \mathbb{F} \to \mathcal{K}$.

Many ways exist to learn classifiers $\kappa$ given training data, i.e. a collection of labeled instances $(\mathbf{v}, c)$, including decision trees [23] and their ensembles [11, 12], decision lists [57], neural networks [35], etc. Hereinafter, this paper considers boosted tree (BT) models trained with the use of XGBoost [12].

**Example 1.** *Figure 1 shows a BT model trained for a simplified version of the* adult *dataset [33]. For a data instance $\mathbf{v} = \{$Education = Bachelors, Status = Separated, Occupation = Sales, Relation-*

---

[1]Any set of classes $\{c_1, \ldots, c_k\}$ can always be mapped into the set of the corresponding indices $\{1, \ldots, k\}$.

*ship = Not-in-family, Sex = Male, Hours/w ≤ 40}, the model predicts <50k because the sum of the weights in the 3 trees for this instance equals* $-0.4073 = (-0.1089 - 0.2404 - 0.0580) < 0$.

## 2.2 ML Model Interpretability and Post-Hoc Explanations

Interpretability is generally accepted to be a subjective concept, without a formal definition [39]. One way to measure interpretability is in terms of the succinctness of information provided by an ML model to justify a given prediction. Recent years have witnessed an upsurge in the interest in devising and applying interpretable models in safety-critical applications [48, 58]. An alternative to interpretable models is post-hoc explanation of *black-box* models, which this paper focuses on.

Numerous methods to compute explanations have been proposed recently [46, 48]. The lion's share of these comprise what is called *model-agnostic* approaches to explainability [40, 55, 56] of heuristic nature that resort to extensive sampling in the vicinity of an instance being explained in order to "estimate" the behavior of the classifier in this local vicinity of the instance. In this regard, they rely on estimating input data distribution by building on the information about the training data [34]. Depending on the form of explanations model-agnostic approaches offer, they are conventionally classified as *feature selection* or *feature attribution* approaches briefly discussed below.

**Feature Selection.** A feature selection approach identifies subsets of features that are deemed *sufficient* for a given prediction $c = \kappa(\mathbf{v})$. As mentioned above, the majority of feature selection approaches are model-agnostic with one prominent example being Anchors [56]. As such, the sufficiency of the selected set of features for a given prediction is determined statistically based on extensive sampling around the instance of interest, by assessing a few measures like *fidelity*, *precision*, among others. As a result, feature selection explanations given as a set of features $\omega \subseteq \mathcal{F}$ should be interpreted as the conjunction $\bigwedge_{i \in \omega} (x_i = v_i)$ deemed responsible for prediction $c = \kappa(\mathbf{v})$, $\mathbf{v} \in \mathbb{F}$, $c \in \mathcal{K}$. Due to the statistical nature of these explainers, they are known to suffer from various explanation quality issues [24, 34, 63]. An additional line of work on *formal* explainability [25, 61] also tackles feature selection while offering guarantees of soundness; these are discussed below.

**Feature Attribution.** A different view on post-hoc explanations is provided by feature attribution approaches, e.g. LIME [55] and SHAP [40]. Based on random sampling in the neighborhood of the target instance, these approaches attribute responsibility to all model's features by assigning a numeric value $w_i \in \mathbb{R}$ of importance to each feature $i \in \mathcal{F}$. Given these importance values, the features can then be ranked from most important to least important. As a result, a feature attribution explanation is conventionally provided as a linear form $\sum_{i \in \mathcal{F}} w_i \cdot x_i$, which can be also seen as approximating the original black-box explainer $\kappa$ in the *local* neighborhood of instance $\mathbf{v} \in \mathbb{F}$. Among other feature attribution approaches, SHAP [5, 6, 40] is often claimed to stand out as it aims at approximating Shapley values, a powerful concept originating from cooperative games in game theory [60].

**Formal Explainability.** In this work, we build on formal explainability proposed in earlier work [8, 13, 25, 42, 61]. where explanations are equated with *abductive explanations* (AXp's). Abductive explanations are *subset-minimal* sets of features formally proved to suffice to explain an ML prediction given a formal representation of the classifier of interest. Concretely, given an instance $\mathbf{v} \in \mathbb{F}$ and a prediction $c = \kappa(\mathbf{v})$, an AXp is a subset-minimal set of features $\mathcal{X} \subseteq \mathcal{F}$, such that

$$\forall (\mathbf{x} \in \mathbb{F}). \bigwedge_{i \in \mathcal{X}} (x_i = v_i) \rightarrow (\kappa(\mathbf{x}) = c) \tag{1}$$

Abductive explanations are guaranteed to be subset-minimal sets of features proved to satisfy (1). As other feature selection explanations, they answer *why* a certain prediction was made. An alternate way to explain a model's behavior is to seek an answer *why not* another prediction was made, or, in other words, *how* to change the prediction. Explanations answering *why not* questions are referred to as *contrastive explanations* (CXp's) [26, 42, 46]. As in prior work, we define a CXp as a subset-minimal set of features that, if allowed to change their values, are *necessary* to change the prediction of the model. Formally, a CXp for prediction $c = \kappa(\mathbf{v})$ is a subset-minimal set of features $\mathcal{Y} \subseteq \mathcal{F}$, such that

$$\exists (\mathbf{x} \in \mathbb{F}). \bigwedge_{i \notin \mathcal{Y}} (x_i = v_i) \wedge (\kappa(\mathbf{x}) \neq c) \tag{2}$$

Finally, recent work has shown that AXp's and CXp's for a given instance $\mathbf{v} \in \mathbb{F}$ are related through the *minimal hitting set duality* [26, 54]. The duality implies that each AXp for a prediction $c = \kappa(\mathbf{v})$

is a *minimal hitting set*[2] (MHS) of the set of all CXp's for that prediction, and the other way around: each CXp is an MHS of the set of all AXp's. The explanation enumeration algorithm [26] applied in this paper heavily relies on this duality relation and is inspired by the MARCO algorithm originating from the area of over-constrained systems [36, 37, 53]. A growing body of recent work on formal explanations is represented (but not limited) by [2, 4, 7, 9, 10, 14, 18, 20, 27, 41–44, 65].

**Example 2.** *In the context of Example 1, feature attribution computed by LIME and SHAP as well as all 2 AXp's are shown in Figure 2. AXp $\mathcal{X}_1$ indicates that specifying Education = Bachelors and Hours/w $\leq$ 40 guarantees that any compatible instance is classified as < 50k independent of the values of other features, e.g. Status and Relationship, since the maximal sum of weights is $0.0770 - 0.0200 - 0.0580 = -0.0010 < 0$ as long as the feature values above are used. Observe that another AXp $\mathcal{X}_2$ for $\mathbf{v}$ is {Education, Status}. Since both of the two AXp's for $\mathbf{v}$ consist of two features, it is difficult to judge which one is better without a formal feature importance assessment.*

## 3    Why Formal Feature Attribution?

On the one hand, abductive explanations serve as a viable alternative to non-formal feature selection approaches because they (i) guarantee subset-minimality of the selected sets of features and (ii) are computed via formal reasoning over the behavior of the corresponding ML model. Having said that, they suffer from a few issues. First, observe that deciding the validity of (1) requires a formal reasoner to take into account the complete feature space $\mathbb{F}$, assuming that the features are independent and uniformly distributed [65]. In other words, the reasoner has to check all the combinations of feature values, including those that *never appear in practice*. This makes AXp's being unnecessarily *conservative* (long), i.e. they may be hard for a human decision maker to interpret. Second, AXp's are not aimed at providing feature attribution. The abundance of various AXp's for a single data instance [25], e.g. see Example 2, exacerbates this issue as it becomes unclear for a user which of the AXp's to use to make an informed decision in a particular situation.

On the other hand, non-formal feature attribution in general is known to be susceptible to out-of-distribution sampling [34, 62] while SHAP is shown to fail to effectively approximate Shapley values [21]. Moreover and quite surprisingly, [21] argued that even the use of exact Shapley values is inadequate as a measure of feature importance. Our results below confirm that both LIME and SHAP often fail to grasp the real feature attribution in a number of practical scenarios.

To address the above limitations, we propose the concept of *formal feature attribution* (FFA) as defined next. Let us denote the set of all formal abductive explanations for a prediction $c = \kappa(\mathbf{v})$ by $\mathbb{A}_\kappa(\mathbf{v}, c)$. Then formal feature attribution of a feature $i \in \mathcal{F}$ can be defined as the proportion of abductive explanations where it occurs. More formally,

**Definition 1: (FFA).** The *formal feature attribution* $\mathrm{ffa}_\kappa(i, (\mathbf{v}, c))$ of a feature $i \in \mathcal{F}$ to an instance $(\mathbf{v}, c)$ for machine learning model $\kappa$ is

$$\mathrm{ffa}_\kappa(i, (\mathbf{v}, c)) = \frac{|\{\mathcal{X} \mid \mathcal{X} \in \mathbb{A}_\kappa(\mathbf{v}, c), i \in \mathcal{X}\}|}{|\mathbb{A}_\kappa(\mathbf{v}, c)|} \tag{3}$$

Formal feature attribution has some nice properties. First, it has a strict and formal definition, i.e. we can, assuming we are able to compute the complete set of AXp's for an instance, exactly define it for all features $i \in \mathcal{F}$. Second, it is fairly easy to explain to a user of the classification system, even if they are non-expert. Namely, it is the percentage of (formal abductive) explanations that make use of a particular feature $i$. Third, as we shall see later, even though we may not be able to compute all AXp's exhaustively, we can still get good approximations fast.

**Example 3.** *Recall Example 2. As there are 2 AXp's for instance $\mathbf{v}$, the prediction can be attributed to the 3 features with non-zero FFA shown in Figure 2d. Also, observe how both LIME and SHAP (see Figure 2a and Figure 2b) assign non-zero attribution to the feature Relationship, which is in fact irrelevant for the prediction, but overlook the highest importance of feature Education.*

One criticism of the above definition is that it does not take into account the length of explanations where the feature arises. Arguably if a feature arises in many AXp's of size 2, it should be considered

---

[2]Given a set of sets $\mathbb{S}$, a *hitting set* of $\mathbb{S}$ is a set $H$ such that $\forall S \in \mathbb{S}, S \cup H \neq \emptyset$, i.e. $H$ "hits" every set in $\mathbb{S}$. A hitting set $H$ for $\mathbb{S}$ is *minimal* if none of its strict subsets is also a hitting set.

more important than a feature which arises in the same number of AXp's but where each is of size 10. An alternate definition, which tries to take this into account, is the weighted formal feature attribution (WFFA), i.e. the *average* proportion of AXp's that include feature $i \in \mathcal{F}$. Formally,

**Definition 2: (WFFA).** The *weighted formal feature attribution* $\mathrm{wffa}_\kappa(i, (\mathbf{v}, c))$ of a feature $i \in \mathcal{F}$ to an instance $(\mathbf{v}, c)$ for machine learning model $\kappa$ is

$$\mathrm{wffa}_\kappa(i, (\mathbf{v}, c)) = \frac{\sum_{\mathcal{X} \in \mathbb{A}_\kappa(\mathbf{v}, c), i \in \mathcal{X}} |\mathcal{X}|^{-1}}{|\mathbb{A}_\kappa(\mathbf{v}, c)|} \tag{4}$$

Note that these attribution values are not on the same scale although they are convertible:

$$\sum_{i \in \mathcal{F}} \mathrm{ffa}_\kappa(i, (\mathbf{v}, c)) = \frac{\sum_{\mathcal{X} \in \mathbb{A}_\kappa(\mathbf{v}, c)} |\mathcal{X}|}{|\mathbb{A}_\kappa(\mathbf{v}, c)|} \times \sum_{i \in \mathcal{F}} \mathrm{wffa}_\kappa(i, (\mathbf{v}, c)).$$

FFA can be related to the problem of *feature relevancy* [22], where a feature is said to be *relevant* if it belongs to at least one AXp. Indeed, feature $i \in \mathcal{F}$ is relevant for prediction $c = \kappa(\mathbf{v})$ if and only if $\mathrm{ffa}_\kappa(i, (\mathbf{v}, c)) > 0$. As a result, the following claim can be made.

**Proposition 1.** *Given a feature $i \in \mathcal{F}$ and a prediction $c = \kappa(\mathbf{v})$, deciding whether $\mathrm{ffa}_\kappa(i, (\mathbf{v}, c)) > \omega$, $\omega \in (0, 1]$, is at least as hard as deciding whether feature $i$ is relevant for the prediction.*

The above result indicates that computing exact FFA values may be expensive in practice. For example and in light of [22], one can conclude that the decision version of the problem is $\Sigma_2^P$-hard in the case of DNF classifiers.

Similarly and using the relation between FFA and feature relevancy above, we can note that the decision version of the problem is in $\Sigma_2^P$ as long as deciding the validity of (1) is in NP, which in general is the case (unless the problem is simpler, e.g. for decision trees [28]). Namely, the following result is a simple consequence of the membership result for the feature relevance problem [22].

**Proposition 2.** *Deciding whether $\mathrm{ffa}_\kappa(i, (\mathbf{v}, c)) > \omega$, $\omega \in (0, 1]$, is in $\Sigma_2^P$ if deciding (1) is in NP.*

## 4   Approximating Formal Feature Attribution

As the previous section argues and as our experimental results confirm, it may be challenging in practice to compute exact FFA values due to the general complexity of the problem. Although some ML models admit efficient formal encodings and reasoning procedures, effective principal methods for FFA approximation seem necessary. This section proposes one such method.

Normally, formal explanation enumeration is done by exploiting the MHS duality between AXp's and CXp's and the use of MARCO-like [37] algorithms aiming at efficient exploration of minimal hitting sets of either AXp's or CXp's [26, 36, 37, 53]. Depending on the target type of formal explanation, MARCO exhaustively enumerates all such explanations one by one, each time extracting a candidate minimal hitting set and checking if it is a desired explanation. If it is then it is recorded and blocked such that this candidate is never repeated again. Otherwise, a dual explanation is extracted from the subset of features complementary to the candidate [25], gets recorded and blocked so that it is hit by each future candidate. The procedure proceeds until no more hitting sets of the set of dual explanations can be extracted, which signifies that all target explanations are enumerated. Observe that while doing so, MARCO also enumerates all the dual explanations as a kind of "side effect".

One of the properties of MARCO used in our approximation approach is that it is an *anytime* algorithm, i.e. we can run it for as long as we need to get a sufficient number of explanations. This means we can stop it by using a timeout or upon collecting a certain number of explanations.

The main insight of FFA approximation is as follows. Recall that to compute FFA, we are interested in AXp enumeration. Although intuitively this suggests the use of MARCO targeting AXp's, for the sake of fast and high-quality FFA approximation, we propose to target CXp enumeration with AXp's as dual explanations computed "unintentionally". The reason for this is twofold: (i) we need to get a good FFA approximation as fast as we can and (ii) according to our practical observations, MARCO needs to amass a large number of dual explanations before it can start producing target explanations. This is because the hitting set enumerator is initially "blind" and knows nothing about the features

**Algorithm 1** MARCO-like Anytime Explanation Enumeration

---

1: **procedure** XPENUM($\kappa$, $\mathbf{v}$, $c$)
2:    $(\mathbb{A}, \mathbb{C}) \leftarrow (\emptyset, \emptyset)$             ▷ *Sets of AXp's and CXp's to collect.*
3:    **while** true **do**
4:       $\mathcal{Y} \leftarrow$ MINIMALHS($\mathbb{A}, \mathbb{C}$)          ▷ *Get a new MHS of $\mathbb{A}$ subject to $\mathbb{C}$.*
5:       **if** $\mathcal{Y} = \bot$ **then break**           ▷ *Stop if none is computed.*
6:       **if** $\exists(\mathbf{x} \in \mathbb{F}). \bigwedge_{i \notin \mathcal{Y}}(x_i = v_i) \wedge (\kappa(\mathbf{x}) \neq c)$ **then**    ▷ *Check CXp condition (2) for $\mathcal{Y}$.*
7:          $\mathbb{C} \leftarrow \mathbb{C} \cup \{\mathcal{Y}\}$           ▷ *$\mathcal{Y}$ appears to be a CXp.*
8:       **else**           ▷ *There must be a missing AXp $\mathcal{X} \subseteq \mathcal{F} \setminus \mathcal{Y}$.*
9:          $\mathcal{X} \leftarrow$ EXTRACTAXP($\mathcal{F} \setminus \mathcal{Y}, \kappa, \mathbf{v}, c$)   ▷ *Get AXp $\mathcal{X}$ by iteratively checking (1) [25].*
10:          $\mathbb{A} \leftarrow \mathbb{A} \cup \{\mathcal{X}\}$           ▷ *Collect new AXp $\mathcal{X}$.*
   **return** $\mathbb{A}, \mathbb{C}$

---

it should pay attention to — it uncovers this information gradually by collecting dual explanations to hit. This way a large number of dual explanations can quickly be enumerated during this initial phase of grasping the search space, essentially "for free". Our experimental results demonstrate the effectiveness of this strategy in terms of monotone convergence of approximate FFA to the exact FFA with the increase of the time limit. A high-level view of the version of MARCO used in our approach targeting CXp enumeration and amassing AXp's as dual explanations is shown in Algorithm 1.

# 5 Experimental Evidence

This section assesses the formal feature attribution for gradient boosted trees (BT) [12] on multiple widely used images and tabular datasets, and compares FFA with LIME and SHAP. In addition, it also demonstrates the use of FFA in a real-world scenario of Just-in-Time (JIT) defect prediction, which assists teams in prioritizing their limited resources on high-risk commits or pull requests [52].

**Setup and Prototype Implementation.** All experiments were performed on an Intel Xeon 8260 CPU running Ubuntu 20.04.2 LTS, with the memory limit of 8 GByte. A prototype of the approach implementing Algorithm 1 and thus producing FFA was developed as a set of Python scripts and builds on [27]. As the FFA and WFFA values turn out to be almost identical (subject to normalization) in our experiments, here we report only FFA. WFFA results can be found in supplementary material.

**Datasets and Machine Learning Models.** The well-known MNIST dataset [15, 50] of hand-written digits 0–9 is considered, with two concrete binary classification tasks created: 1 vs. 3 and 1 vs. 7. We also consider PneumoniaMNIST [67], a binary classification dataset to distinguish X-ray images of pneumonia from normal cases. To demonstrate extraction of *exact* FFA values for the above datasets, we also examine their downscaled versions, i.e. reduced from $28 \times 28 \times 1$ to $10 \times 10 \times 1$. We also consider 11 tabular datasets often applied in the area of ML explainability and fairness [3, 16, 17, 19, 49, 59]. All the considered datasets are randomly split into 80% training and and 20% test data. For images, 15 test instances are randomly selected in each test set for explanation while all tabular test instances are explained. For all datasets, gradient boosted trees (BTs) are trained by XGBoost [12], where each BT consists of 25 trees of depth 3 per class.[3] Finally, we show the use of FFA on 2 JIT defect prediction datasets [52], with 500 instances per dataset chosen for analysis.

## 5.1 Formal Feature Attribution

In this section, we restrict ourselves to examples where we can compute the *exact* FFA values for explanations by computing all AXp's. To compare with LIME and SHAP, we take their solutions, replace negative attributions by the positive counterpart (in a sense taking the absolute value) and then normalize the values into $[0, 1]$. We then compare these approaches with the computed FFA values, which are also in $[0, 1]$. The *error* is measured as Manhattan distance, i.e. the sum of absolute differences across all features. We also compare feature rankings according to the competitors (again using absolute values for LIME and SHAP) using Kendall's Tau [31] and rank-biased overlap (RBO) [66]

---

[3]Test accuracy for MNIST digits is 0.99, while it is 0.83 for PneumoniaMNIST. This holds both for the $28 \times 28$ and $10 \times 10$ versions of the datasets. The average accuracy across the 11 selected tabular datasets is 0.80.

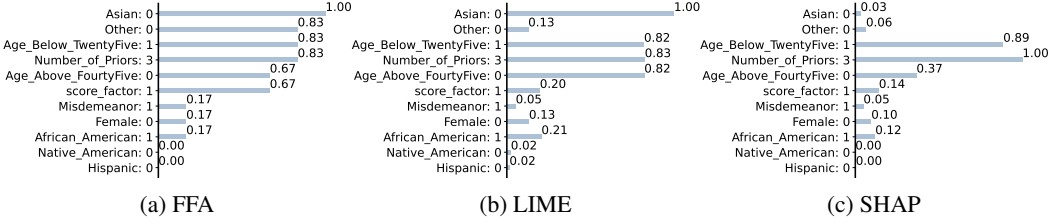

|              | (a) FFA | (b) LIME | (c) SHAP |
|---|---|---|---|

Figure 3: Explanations for an instance of Compas $\mathbf{v} = \{\#\text{Priors} = 3, \text{Score\_factor} = 1, \text{Age\_Above\_FourtyFive} = 0, \text{Age\_Below\_TwentyFive} = 1, \text{African\_American} = 1, \text{Asian} = 0, \text{Hispanic} = 0, \text{Native\_American} = 0, \text{Other} = 0, \text{Female} = 0, \text{Misdemeanor} = 1\}$ predicted as Two_yr_Recidivism = true.

Table 1: LIME and SHAP versus FFA on tabular data.

| Dataset ($|\mathcal{F}|$) | adult (12) | appendicitis (7) | australian (14) | cars (8) | compas (11) | heart-statlog (13) | hungarian (13) | lending (9) | liver-disorder (6) | pima (8) | recidivism (15) |
|---|---|---|---|---|---|---|---|---|---|---|---|
| **Approach** | | | | | | **Error** | | | | | |
| LIME | 4.48 | 2.25 | 5.13 | 1.53 | 3.28 | 4.48 | 4.56 | 1.39 | 2.39 | 2.72 | 4.73 |
| SHAP | 4.47 | 2.01 | 4.49 | 1.40 | 2.67 | 3.71 | 4.14 | 1.44 | 2.28 | 3.00 | 4.76 |
| | | | | | | **Kendall's Tau** | | | | | |
| LIME | 0.07 | 0.11 | 0.22 | -0.11 | -0.11 | 0.17 | 0.04 | -0.36 | -0.22 | 0.17 | 0.05 |
| SHAP | 0.03 | 0.12 | 0.27 | -0.10 | -0.10 | 0.17 | 0.20 | -0.39 | -0.21 | 0.07 | 0.12 |
| | | | | | | **RBO** | | | | | |
| LIME | 0.54 | 0.66 | 0.49 | 0.63 | 0.55 | 0.56 | 0.41 | 0.59 | 0.66 | 0.68 | 0.39 |
| SHAP | 0.49 | 0.67 | 0.55 | 0.66 | 0.59 | 0.52 | 0.49 | 0.61 | 0.67 | 0.63 | 0.44 |

metrics.[4] Kendall's Tau and RBO are measured on a scale $[-1, 1]$ and $[0, 1]$, respectively. A higher value in both metrics indicates better agreement or closeness between a ranking and FFA.

**Tabular Data.** Figure 3 exemplifies a comparison of FFA, LIME and SHAP on an instance of the Compas dataset [3]. While FFA and LIME agree on the most important feature, "Asian", SHAP gives it very little weight. Neither LIME nor SHAP agree with FFA, though there is clearly some similarity.

Table 1 details the comparison conducted on 11 tabular datasets, including *adult*, *compas*, and *recidivism* datasets commonly used in XAI. For each dataset, we calculate the metric for each individual instance and then average the outcomes to obtain the final result for that dataset. As can be observed, the errors of LIME's feature attribution across these datasets span from 1.39 to 5.13. SHAP demonstrates similar errors within a range $[1.40, 4.76]$. LIME and SHAP also exhibit comparable performance in relation to the two ranking comparison metrics. The values of Kendall's Tau for LIME (resp. SHAP) are between $-0.36$ and $0.22$ (resp. $-0.39$ and $0.27$). Regarding the RBO values, LIME exhibits values between 0.39 and 0.68, whereas SHAP demonstrates values ranging from 0.44 to 0.67. Overall, as Table 1 indicates, both LIME and SHAP fail to get close enough to FFA.

**$10 \times 10$ Digits.** We now compare the results on $10 \times 10$ downscaled MNIST digits and PneumoniaMNIST images, where it is feasible to compute all AXp's. Table 2 compares LIME's, SHAP's feature attribution and approximate FFA. Here, we run AXp enumeration for a number of seconds, which is denoted as $\text{FFA}_*, * \in \mathbb{R}^+$. The runtime required for each image by LIME and SHAP is less than one second. The results show that the errors of our approximation are small, even after 10 seconds it beats both LIME and SHAP, and decreases as we generate more AXp's. The results for the orderings show again that after 10 seconds, $\text{FFA}_*$ ordering gets closer to the exact FFA than both LIME and SHAP. Observe how LIME is particularly far away from the *exact* FFA ordering.

**Summary.** *These results make us confident that we can get useful approximations to the exact FFA without exhaustively computing all AXp's while feature attribution determined by LIME and SHAP is quite erroneous and fails to provide a human-decision maker with useful insights, despite being fast.*

---

[4]Kendall's Tau is a correlation coefficient assessing the ordinal association between two ranked lists, offering a measure of similarity in the order of values; on the other hand, RBO is a metric that measures the similarity between two ranked lists, taking into account both the order and the depth of the overlap.

Table 2: Comparison on $10 \times 10$ Images of FFA versus LIME, SHAP and FFA approximations.

| Dataset | LIME | SHAP | FFA$_{10}$ | FFA$_{30}$ | FFA$_{60}$ | FFA$_{120}$ | FFA$_{600}$ | FFA$_{1200}$ |
|---|---|---|---|---|---|---|---|---|
| $(|\mathcal{F}| = 100)$ | | | | | Error | | | |
| 10×10-mnist-1vs3 | 11.50 | 10.07 | 5.74 | 5.33 | 4.97 | 4.62 | 3.37 | 2.67 |
| 10×10-mnist-1vs7 | 12.64 | 8.28 | 4.16 | 3.58 | 2.94 | 2.50 | 1.42 | 1.01 |
| 10×10-pneumoniamnist | 17.32 | 17.90 | 5.37 | 4.32 | 3.78 | 3.39 | 2.22 | 1.64 |
| | | | | | Kendall's Tau | | | |
| 10×10-mnist-1vs3 | -0.15 | 0.48 | 0.49 | 0.57 | 0.62 | 0.65 | 0.74 | 0.80 |
| 10×10-mnist-1vs7 | -0.33 | 0.47 | 0.52 | 0.63 | 0.70 | 0.77 | 0.85 | 0.89 |
| 10×10-pneumoniamnist | -0.02 | 0.24 | 0.58 | 0.71 | 0.79 | 0.80 | 0.89 | 0.92 |
| | | | | | RBO | | | |
| 10×10-mnist-1vs3 | 0.20 | 0.50 | 0.61 | 0.65 | 0.69 | 0.74 | 0.81 | 0.84 |
| 10×10-mnist-1vs7 | 0.19 | 0.58 | 0.73 | 0.77 | 0.81 | 0.86 | 0.90 | 0.90 |
| 10×10-pneumoniamnist | 0.21 | 0.37 | 0.61 | 0.70 | 0.73 | 0.77 | 0.83 | 0.87 |

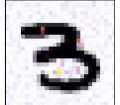 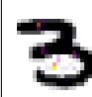 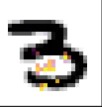 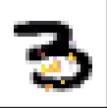 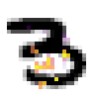 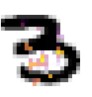 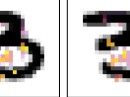 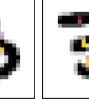

(a) LIME   (b) SHAP   (c) FFA$_{10}$   (d) FFA$_{30}$   (e) FFA$_{120}$   (f) FFA$_{600}$   (g) FFA$_{1.2k}$   (h) FFA$_{3.6k}$   (i) FFA$_{7.2k}$

Figure 4: $28 \times 28$ MNIST 1 vs. 3. The prediction is digit 3. The *plasma* gradient is used ranging from deep purple for the least important features to vibrant yellow for the most important features.

Table 3: Comparison on $28 \times 28$ Images of FFA$_{7200}$ versus LIME, SHAP and FFA approximations.

| Dataset | LIME | SHAP | FFA$_{10}$ | FFA$_{30}$ | FFA$_{120}$ | FFA$_{600}$ | FFA$_{1200}$ | FFA$_{3600}$ |
|---|---|---|---|---|---|---|---|---|
| $(|\mathcal{F}| = 784)$ | | | | | Error | | | |
| 28×28-mnist-1vs3 | 49.66 | 22.77 | 9.44 | 7.61 | 6.81 | 4.51 | 3.13 | 2.69 |
| 28×28-mnist-1vs7 | 55.10 | 24.92 | 11.78 | 9.58 | 6.94 | 4.51 | 3.30 | 2.18 |
| 28×28-pneumoniamnist | 62.94 | 31.55 | 8.17 | 7.81 | 5.69 | 4.89 | 3.77 | 3.10 |
| | | | | | Kendall's Tau | | | |
| 28×28-mnist-1vs3 | -0.80 | 0.42 | 0.44 | 0.62 | 0.69 | 0.80 | 0.86 | 0.87 |
| 28×28-mnist-1vs7 | -0.79 | 0.34 | 0.40 | 0.56 | 0.72 | 0.82 | 0.87 | 0.92 |
| 28×28-pneumoniamnist | -0.66 | 0.24 | 0.34 | 0.50 | 0.67 | 0.76 | 0.80 | 0.87 |
| | | | | | RBO | | | |
| 28×28-mnist-1vs3 | 0.03 | 0.40 | 0.43 | 0.50 | 0.61 | 0.78 | 0.83 | 0.88 |
| 28×28-mnist-1vs7 | 0.03 | 0.34 | 0.40 | 0.45 | 0.58 | 0.76 | 0.83 | 0.93 |
| 28×28-pneumoniamnist | 0.03 | 0.23 | 0.31 | 0.35 | 0.42 | 0.59 | 0.66 | 0.83 |

## 5.2 Approximating Formal Feature Attribution

Since the problem of formal feature attribution "lives" in $\Sigma_2^P$, it is not surprising that computing FFA may be challenging in practice. Table 2 suggests that our approach gets good FFA approximations even if we only collect AXp's for a short time. Here we compare the fidelity of our approach versus the approximate FFA computed after 2 hours (7200s). Figure 4, 5, and 6 depict feature attributions generated by LIME, SHAP and FFA$_*$ for the three selected $28 \times 28$ images. The comparison between LIME, SHAP, and the approximate FFA computation is detailed in Table 3. The LIME and SHAP processing time for each image is less than one second. The average findings detailed in Table 3 are consistent with those shown in Table 2. Namely, FFA approximation yields better errors, Kendall's Tau and RBO values, outperforming both LIME, and SHAP after 10 seconds. Furthermore, the results demonstrate that after 10 seconds our approach places feature attributions closer to FFA$_{7200}$ compared to both LIME and SHAP hinting on the features that are truly relevant for the prediction.

## 5.3 Application in Just-in-Time Defect Prediction

Just-in-Time (JIT) defect prediction [30, 32, 38, 51] has been recently proposed to predict if a commit will introduce software defects in the future, enabling development teams to prioritize their limited Software Quality Assurance resources on the most risky commits/pull requests. The approach of JIT

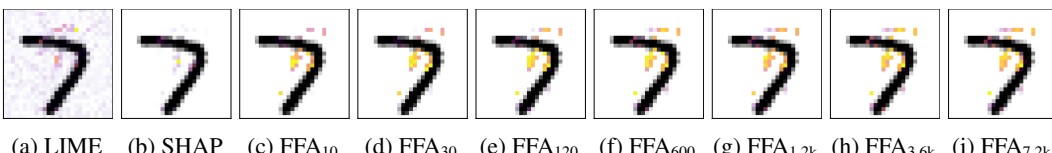

(a) LIME    (b) SHAP    (c) FFA$_{10}$    (d) FFA$_{30}$    (e) FFA$_{120}$    (f) FFA$_{600}$    (g) FFA$_{1.2k}$    (h) FFA$_{3.6k}$    (i) FFA$_{7.2k}$

Figure 5: $28 \times 28$ MNIST 1 vs. 7. The prediction is digit 7.

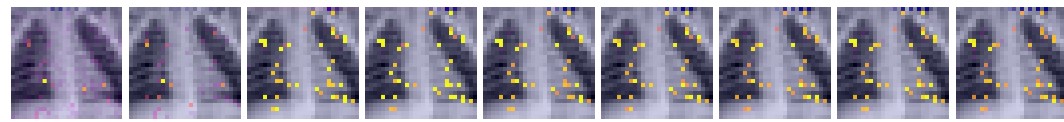

(a) LIME    (b) SHAP    (c) FFA$_{10}$    (d) FFA$_{30}$    (e) FFA$_{120}$    (f) FFA$_{600}$    (g) FFA$_{1.2k}$    (h) FFA$_{3.6k}$    (i) FFA$_{7.2k}$

Figure 6: $28 \times 28$ PneumoniaMNIST. The prediction is normal.

Table 4: Just-in-Time Defect Prediction comparison of FFA versus LIME and SHAP.

| Approach | openstack ($|\mathcal{F}| = 13$) | | | qt ($|\mathcal{F}| = 16$) | | |
|---|---|---|---|---|---|---|
| | Error | Kendall's Tau | RBO | Error | Kendall's Tau | RBO |
| LIME | 4.84 | 0.05 | 0.55 | 5.63 | -0.08 | 0.45 |
| SHAP | 5.08 | 0.00 | 0.53 | 5.22 | -0.13 | 0.44 |

defect prediction has often been considered a black-box, lacking explainability for practitioners. To tackle this challenge, our proposed approach to generating FFA can be employed, as model-agnostic approaches cannot guarantee to provide accurate feature attribution (see above). We use logistic regression models of [52] based on large-scale open-source Openstack and Qt datasets provided by [45] commonly used for JIT defect prediction [52]. Monotonicity of logistic regression enables us to enumerate explanations using the approach of [44] and so to extract *exact FFA* for each instance *within a second*. Table 4 details the comparison of FFA, LIME and SHAP in terms of the three considered metrics. As with the outcomes presented in Table 1, Table 2, and Table 3, neither LIME nor SHAP align with formal feature attribution, though there are some similarities between them.

## 6   Limitations

Despite the rigorous guarantees provided by formal feature attribution and high-quality of the result explanations, the following limitations can be identified. First, our approach relies on formal reasoning and thus requires an ML model of interest to admit a representation in some fragments of first-order logic, and the corresponding reasoner to deal with it [42]. Second, the problem complexity impedes immediate and widespread use of FFA and signifies the need to develop effective methods of FFA approximation. Finally, though our experimental evidence suggests that FFA approximations quickly converge to the exact values of FFA, whether or not this holds in general remains an open question.

## 7   Conclusions

Most approaches to XAI are heuristic methods that are susceptible to unsoundness and out-of-distribution sampling. Formal approaches to XAI have so far concentrated on the problem of feature selection, detecting which features are important for justifying a classification decision, and not on feature attribution, where we can understand the weight of a feature in making such a decision. In this paper we define the first formal approach to feature attribution (FFA) we are aware of, using the proportion of abductive explanations in which a feature occurs to weight its importance. We show that we can compute FFA exactly for many classification problems, and when we cannot we can compute effective approximations. Existing heuristic approaches to feature attribution do not agree with FFA. Sometimes they markedly differ, for example, assigning no weight to a feature that appears in (a large number of) explanations, or assigning (large) non-zero weight to a feature that is irrelevant for the prediction. Overall, the paper argues that if we agree that FFA is a correct measure of feature attribution then we need to investigate methods that compute good FFA approximations quickly.

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
