# OpenReview forum: "On Formal Feature Attribution and Its Approximation"
_NeurIPS.cc/2023/Conference — Submitted to NeurIPS 2023_

### Official Review · Reviewer_CksZ · 2023-07-04

**Soundness:** 2 fair
**Presentation:** 4 excellent
**Contribution:** 2 fair
**Rating:** 4
**Confidence:** 4

**Summary:**

The paper proposes and studies a notion of feature attribution in which features are scored for a given instance according to the proportion of minimal explanations for that instance in which they participate. Although an exact computation of this scores can be computationally unfeasible, the paper exploits the minimal-hitting-set duality between abductive and contrastive explanations to approximate them efficiently. Finally, they study their approach empirically over several datasets.

**Strengths:**

- The paper is really well written and easy to follow; presentation is excellent
- The paper does a good job at showcasing the importance of the problem and providing references to issues that are present in SHAP
- The proposed metric is simple and seems promising
- The experiments have good results and are presented nicely
- The sections on limitations and conclusions provide a nice and helpful starting point for further discussion
- References seem appropriate and detailed

**Weaknesses:**

- The paper does not provide proofs in the supplementary material for its propositions. While proposition 1 is self-explanatory, proposition 2 is not and should be accompanied with a proof.

- Despite mentioning the weighted variant (Definition 2),  the paper doesn't seem to do anything with it; there are no theoretical nor practical results about it as far as I can see, and it is only discussed in the appendix. I understand that due to the page limit not everything can fit in the paper, but including the definition of WFFA without doing anything with it in the paper seems like a poor choice to me.

- Even though the paper is about formal explainability, and the scores themselves are formally defined, the approximation algorithm doesn't seem to have any formal guarantee, and thus it is unclear to me what the interpretation of the results should be. It seems to me that further theoretical studies are required; how do we know that their approach can't fall into cases where it gives answers that are arbitrarily far from the ground truth? The starting point of the paper is about how methods such as SHAP have pitfalls when certain conditions are met, and yet it is not clear at all that this approach is exempt of similar (or even worse) potential problems.

- The experimental data is, unless I am missing something, a bit strange; LIME, SHAP and their FFA approximation compute different things, both in theory and practice, and thus I don't understand at all what the meaning of their comparison is; LIME and SHAP are not approximations to the FFA score, and thus it doesn't feel sound or fair to use them as such. Their idea of taking absolute values and normalizing might make sense, but this is far from obvious and limits how convincing their results are.

**Questions:**

Although I don't have particular questions to the authors,  my perspective (and therefore scores) could change by them convincingly addressing the points raised in the weaknesses section above.

**Limitations:**

Yes; there doesn't seem to be much to address and the authors discuss appropriate points in section 6.

---

> ### Author Rebuttal · Authors · 2023-08-09
>
> Thank you for the comments! In the following, we do our best
> addressing them, which hopefully convinces you that our work has
> enough merits to get accepted.
>
> ### On Missing Proofs
>
> Note that the proof of Proposition 2 is really only a single line,
> making use of results in the cited work [22]:
>
> **Proof of Proposition 2.** From [22] (Proposition 7), deciding whether a
> feature is relevant is in $\Sigma_2^P$ assuming deciding (1) is in NP.
> By Proposition 1, so also is deciding whether $\text{ffa}_\kappa(i,
> (w,c)) > w$.
>
> ### On Weighted FFA
>
> We added the weighted variant because it seems more *"natural"* and
> *"fair"* to treat features in short explanations as having more
> weight. But in practice since the lengths of AXp's are tightly
> clustered around the mean, the unweighted and weighted variants are
> almost indistinguishable. Since the unweighted version is easier to
> explain/understand, we concentrate on this in the paper. However, if
> we omitted the discussion of the weighted version entirely, we believe
> it would be an obvious question for readers, hence we think it is
> worth including.
>
> ### On FFA Approximation Guarantees
>
> First, FFA approximation has one strong property (which we should have
> put in the original paper): Any feature given non-zero importance by
> the approximation, is guaranteed to have non-zero (true/exact)
> importance. Note that this is not the case for LIME and SHAP.
>
> Second, we can argue about the worst-case approximations convergence
> to the true FFA. Suppose we have discovered $n$ out of $N$ total
> AXp's. The FFA approximation for feature $f$ is given by $\hat{p}_f =
> \frac{c_f}{n}$ where $c_f$ is the count of the number of AXp's found
> so far including feature $f$. We have that
>
> $$
> \frac{c_f}{N} \leq \hat{p}_f = \frac{c_f}{n} \leq \frac{c_f + (N-n)}{N}
> $$
>
> The left holds since $n \leq N$ and the right holds since $c_f \leq
> n$. Notice that the true proportion $p_f = \frac{c_f + c'_f}{N}$ also
> lies between these bounds, since $c'_f$, the count of feature $f$ in
> the *remaining* unseen AXp's, is between 0 and $N - n$. Hence the
> error $|p_f - \hat{p}_f|$ is bounded by the max distance from $p_f$ to
> the upper or lower bound. As the number of trials increase, this max
> error decreases. For example, if we have determined half the AXp's, the
> difference between the estimated and the true value is at most half in
> the worst-case scenario.
>
> Of course in practice the error is much lower. This is because in
> practice the AXp's generated to support a single CXp must necessarily
> be *diverse*, which means the error reduces rapidly. Getting a better
> theoretical bound for this behaviour remains (challenging) future
> work.
>
> ### On Experimental Results
>
> Since LIME and SHAP have different ideas of what feature importance
> means, they clearly are not aimed at computing FFA. We compare with
> them to show that these definitions are not the same in practice as
> FFA. In this regard, please also see the general comments above.
> Finally, we normalize results to avoid any differences *simply arising
> from scaling*.

---

> > ### Comment · Reviewer_CksZ · 2023-08-10
> >
> > The response says "Note that the proof of Proposition 2 is really only a single line, making use of results in the cited work [22]:", and then:
> >
> > "Proof of Proposition 2. From [22] (Proposition 7), deciding whether a feature is relevant is in $\Sigma_2^p$
> >  assuming deciding (1) is in NP. By Proposition 1, so also is deciding whether the ffa score is greater-equal than a threshold w".
> >
> > Sorry but this proof strikes as fully wrong, unless I'm missing something important. The fact that deciding feature relevancy is in $\Sigma_2^p$ implies that deciding whether the score is 0 or greater than 0 is in $\Sigma_2^p$, but this implies nothing about whether the problem of deciding whether is greater-equal than, say, $0.1$ is in $\Sigma_2^p$. For instance, if we define the score of a CNF formula as the number of satisfying assignments / total assignments, then clearly deciding "score > 0" is in NP, but deciding whether it's " >= w" for an input "w" would require counting, and unless #P is contained in the polynomial hierarchy, we don't expect the "score >= w" problem to be.
> >
> > Again, I could be misunderstanding something, in which case I'd apologize for the confusion, but I still don't see at all how the problem is in $\Sigma_2^p$.
> >
> > Moreover, the statement of Proposition 1 mentions $w > 0$, which shouldn't be a big issue as the input in which one sets $w = 2^{-m}$ is enough to prove hardness, but I'd have appreciated if both Propositions 1 and 2 were to make explicit that they're considering $w$ as part of the input, and not that there is a fixed $w$ for which the hardness occurs.

---

> > > ### Comment · Reviewer_CksZ · 2023-08-10
> > >
> > > With respect to the approximation guarantee, I think I wasn't fully clear in my original comment; I agree that the quality of the score is basically the same as the fraction of AXps extracted, I wasn't concerned about that part, but rather as to the trade-off between the time required to obtain a good fraction of the AXps. Lines 197-199 say that one can set a timeout, for example, but a timeout doesn't seem to come with a guarantee of number or proportion of AXps extracted, at least by default, thus implying no guarantee on approximation of the score. In other words, I think for the approach to have a guaranteed approximation one would like an algorithm such that, by running it for  $f( |\text{input}|, \epsilon)$ time, for some user-input value $\epsilon$ the proportion of AXps extracted has an approximation guarantee to the true number that's also a function of $\epsilon$.

---

> > > > ### Author Response · Authors · 2023-08-11
> > > > **Time-based approximation guarantees**
> > > >
> > > > You are right that there is no time-based guarantee because there is no way to
> > > > know how many AXp's we can extract within a given time limit. We should say
> > > > this applies in general to any algorithm, including the (approximate)
> > > > computation of Shapley values by SHAP/DeepSHAP/FastSHAP/etc. Given the
> > > > difficulty of the problem, we expect a user to apply a reasonable time limit.
> > > > The experiments show that 10 seconds are enough to get good approximate FFA
> > > > values for the data and models we consider. But if AXp extraction is slower
> > > > for other models and data then 10 seconds will not be sufficient.

---

> > > ### Author Response · Authors · 2023-08-11
> > > **Ambiguity in Proposition 2**
> > >
> > > Thank you for the valuable comment.
> > >
> > > Proposition 1 holds for any fixed value of $\omega>0$.
> > >
> > > Regarding Proposition 2, the confusion comes from the ambiguity regarding the
> > > formulation and quantification of $\omega$, which we apologise for. Proposition 2 should have been stated as follows: "Deciding whether there exists $\omega\in(0,1]$ such that $\text{ffa}(i, (\mathbf{v}, c)) \geq \omega$ is in $\Sigma_2^P$ if deciding (1) is in NP". This essentially means answering whether a given feature's FFA is positive. This holds because we can answer "yes, such an $\omega$ exists" if the feature is relevant; and answer "no such constant $\omega$ exists" if the feature is irrelevant. We will restate the proposition in the final version by removing $\omega$ and instead asking whether $\text{ffa}(i, (\mathbf{v}, c))>0$.
> > >
> > > Again, thank you!

---

### Official Review · Reviewer_FoWS · 2023-07-04

**Soundness:** 3 good
**Presentation:** 4 excellent
**Contribution:** 3 good
**Rating:** 7
**Confidence:** 4

**Summary:**

This paper introduces a novel approach to XAI called Formal Feature Attribution. The authors address the limitations of existing model-agnostic methods and formal XAI approaches by proposing FFA as a solution for feature attribution.
The FFA method leverages formal explanation enumeration to define feature attribution as the proportion of explanations in which a specific feature occurs. The paper highlights the challenges in computing exact FFA but presents an efficient approximation technique using a dual trait of such explanations.

**Strengths:**

The paper exhibits a high level of clarity and coherence in its writing style, making it easily understandable for readers. The claims and contributions are clearly stated, enabling the reader to grasp the main objectives of the research.

The paper makes a noteworthy contribution to the field of Explainable AI by introducing a novel approach. This fills a gap in the existing XAI landscape, where formal foundations are relatively loose, and offers a promising solution for feature attribution.

The authors demonstrate extensive work in various aspects of the research. They have invested effort in both the theoretical aspects, establishing formal foundations for FFA, as well as in the practical aspects, conducting experiments to validate the proposed approach. This comprehensive approach enhances the credibility and reliability of the findings.

**Weaknesses:**

Axiomatic Analysis of FFA: While Lime and Shap have established a set of axioms for explanations, it remains unclear which set of axioms the FFA explanations adhere to. It would be beneficial to explore and define the axioms underlying FFA explanations or engage in theoretical debates surrounding them. For example, I think that duplicate features increase the importance of other features that exists in a common AX'p. Using a specific example, think of a scenario where two correlated features, f1 and f2, are considered. In this case, a decision tree is constructed where f1 appears at a certain node while f2 is absent. Now, create a separate decision tree that includes the condition for either f1 or f2 to appear at that node, resulting in identical predictions due to their correlation. However, despite the consistent predictions, I'm pretty sure that the two models provide different explanations for the features, questioning whether the ratio of feature occurrence in formal explanations increases equally in the numerator and denominator.

Approximation Guarantees for FFA: Since FFA is an NP problem, it necessitates further consideration regarding the guarantees provided by approximation techniques. Investigating the quality and limitations of these approximations would strengthen the practical utility of FFA.

The absence of code implementation limits the reproducibility and practical adoption of the proposed FFA approach.

Addressing these aspects would further enhance the theoretical and practical implications of FFA within the field of Explainable AI.

**Questions:**

Isn't the approach of finding a minimal subset of features for which the classification remains the same too strict?
Could a probabilistic approach potentially be more suitable, enabling the smoothing of results and accommodating variations in feature importance?

**Limitations:**

Distinguishing Out-of-Distribution and Manifold Sampling: The paper does not explicitly discuss the distinctions between out-of-distribution sampling and manifold sampling, despite their potential significance in this context. Exploring the differences and implications of these sampling techniques would contribute to a more comprehensive understanding of FFA.

Addressing Local Explanations: The paper briefly touches on the issue of local explanations, but does not delve into their meanings, differences, or implications. Future research could focus on formulating FFA specifically for global explanations, shedding light on the disparities between local and global explanations within the context of FFA.

---

> ### Author Rebuttal · Authors · 2023-08-09
>
> Thank you for the positive view on our work! Please find our response
> in regard to the weaknesses and limitations you identified in it.
>
> ### On Axiomatic Analysis and Approximation Guarantees
>
> We agree that some initial axiomatic analysis would be nice to have
> for FFA and we will add a couple of comments on this in the final
> version of the paper. For example, we can surely claim that a feature
> irrelevant for a given prediction made by a given model will have a
> zero FFA. Note that this applies both to the exact FFA but also to
> approximate FFA, e.g. taken after a time limit is reached. This is a
> strong result, which does not apply to LIME and SHAP. *Please also see
> a larger comment on FFA approximation guarantees given to reviewer
> CksZ.*
>
> ### On Correlated Features
>
> In your example with two decision trees with correlated features $f_1$
> and $f_2$, the values of FFA will depend on what formal abductive
> explanations are for the two trees. If both trees compute the same
> classification function then the sets of all AXp's will be identical,
> which in turn means the values of FFA will be the same for all the
> features. Otherwise, if the trees have different sets of AXp's, the
> values of FFA for $f_1$ and $f_2$ may indeed differ. We should note that
> there is no issue with this per se because an explainer is meant to
> provide reasons for the behavior of a given classifier in a particular
> situation rather than explain the ground truth. Therefore, the values
> of FFA will be correct for both decision trees as they reflect the
> behavior of these concrete classifiers. Note that we can then use
> these values to compare the classifiers and see which of them is more
> reasonable to apply. Having said that, we agree that the feature
> correlation is a potentially crucial problem that occurs for many
> explanation approaches. From the perspective of formal explanations,
> one can integrate ground truth constraints detecting feature
> correlations when enumerating AXp's [20, *], which will result in more
> reliable FFA values.
>
> [*] Jinqiang Yu, Alexey Ignatiev, Peter J. Stuckey, Nina Narodytska,
> João Marques-Silva: Eliminating the Impossible, Whatever Remains Must
> Be True: On Extracting and Applying Background Knowledge in the
> Context of Formal Explanations. AAAI 2023: 4123-4131
>
> ### On Missing Implementation
>
> We are sorry to see that you could not find the source code of the
> implementation. Please notice that the source code of the
> implementation as well as the complete experimental setup *were
> submitted* with the paper as "supplementary material".
>
> ### On Extracting Subset-Minimal AXp's
>
> You are right that extracting subset-minimal AXp's is computationally
> quite challenging and one could try to accommodate the use of
> probabilistic approaches here. We will consider this in the future and
> will comment on this in the final version.
>
> ### Out-of-Distribution vs Manifold Sampling
>
> Note that as formal explanation approaches (including ours) do not
> rely on sampling, they are not susceptible to *any* sampling issues.
> Having said that, we will surely consider other sampling-based
> explainers in our future work, to see how their feature attribution
> correlates with FFA and how sampling may affect the quality of the
> result explanations.
>
> ### On Local vs Global FFA
>
> Thank you for the comment! We agree this would be a very interesting
> line of future work comparing the pros and cons of local vs global FFA
> and the computational challenges arising in both cases.

---

> > ### Comment · Reviewer_FoWS · 2023-08-12
> >
> > I appreciate your acknowledgment of the clarification regarding the method's independence from sampling. Notably, this aspect has been consistently commented on across all the reviews. I believe it would be beneficial to provide an extended explanation within the paper to address this point more comprehensively.
> >
> > Regarding correlative features, comparing models using scores is possible, but what exactly does this score difference means? Additional elaboration on this aspect [see ref attached] would be valuable.
> >
> > Chen, Hugh, et al. "True to the model or true to the data?." arXiv preprint arXiv:2006.16234 (2020).

---

> > > ### Author Response · Authors · 2023-08-16
> > > **On Additional Comments of Reviewer FoWS**
> > >
> > > Indeed, given the reviewers' confusion, we will add a clarification on the
> > > independence of our method from sampling in the final version of the paper by
> > > using the comments provided in the rebuttal and sacrificing some of the
> > > experimental details in the paper. Thank you for the suggestion.
> > >
> > > On what the score comparison shows, we should once again note that although
> > > attribution scores computed by LIME and SHAP certainly do not target to
> > > replicate FFA, they are still claimed to compute some attribution scores and
> > > the score difference we observe demonstrates that the attribution reported by
> > > these explainers is clearly far away from FFA, a novel, simple, formal, and
> > > easy-to-understand attribution measure. The same observations can be made wrt.
> > > FastSHAP and KernelSHAP additionally tested upon one of the reviewers'
> > > request. We believe these observations should motivate the community to think
> > > over what these explainers actually compute in practice.
> > >
> > > As for correlative features, thank you for the pointer. We will cite this
> > > work. Our method aims at explaining the behavior of the model and so the
> > > explanations we compute are "true to the model". Also note that our method
> > > surely avoids assigning non-zero attribution to the features unused by the
> > > model, i.e. it respects the Dummy Axiom. As for the "interventinal conditional
> > > expectation" studied in this paper and the use of
> > > LIME/SHAP/FastSHAP/KernelSHAP, note that although our running example model
> > > *does use* feature "Relationship", it still *has nothing* to do with the
> > > prediction for the concrete instance discussed in the paper (because this
> > > feature does not belong to any AXp) and so its attribution should be zero
> > > while LIME, SHAP, and KernelSHAP claim it is not. It might be the case that
> > > the issue pertains to the exact Shapley value for this feature in the instance
> > > too (although we have not calculated it). Testing this would be interesting in
> > > the future (to confirm whether the findings of [21] hold here).

---

### Official Review · Reviewer_gWVL · 2023-07-05

**Soundness:** 3 good
**Presentation:** 3 good
**Contribution:** 2 fair
**Rating:** 6
**Confidence:** 4

**Summary:**

The authors propose formal feature attributions, a novel type of local feature attribution method for
explaining the predictions of black box models.  Their approach builds on the notion of abductive explanations (AXp's), a type of minimal sufficient subsets.  One issue with AXps is that there are a potentially exponential number of them, most of which fall outside of the data distribution.  The authors propose - essentially - to summarize the set of AXps via averaging into a per-feature relevance score akin to those provided by LIME and SHAP.  An inverse proposeity weighted variant is also introduced.  The authors then show that typically computing FFAs is computationally intractable
and propose an approximation algorithm for quickly estimating approximate FFAs.  This approximation is shown to work well on two MNIST-like data sets.

**Post-rebuttal update**: increased score, see the discussion.

**Strengths:**

+ Very clearly written, a pleasure to read.
+ Ideas are clearly presented, with a couple of exceptions (see questions below).
+ Related work is well done.
+ FFAs are rooted on a simple and clear concept.
+ Algorithm is sensible.
+ Good empirical performance on a relatively varied set of data sets (for boosted DTs only).
+ Some essential limitations are clearly discussed, but see below.

**Weaknesses:**

- The authors seem to assume FFAs are the "real feature attribution", but provide no real motivation for this (see Q1, this is the big one).
- Unclear reasons why OOD sampling is an issue and how FFAs deal with it (Q2).
- Missing discussion of information loss due to averaging over AXp's (Q3).
- Experiments consider boosted DTs only (relatively minor).
- Missing evaluation of approximate FFA algorithm on non-MNIST data (minor).

**Questions:**

Q1. I am puzzled by the statement in line 144 that "LIME and SHAP often fail to grasp the real feature attribution in a number of practical scenarios", and by the construction of the first experiment.  Here the authors compare LIME and SHAP against exact FFA relevance values, using the term "error" to indicate the difference between relevances output by the former and the latter.  It seems to me the authors automatically assume that Eq. 3 is the "real feature attribution", but it is not clear why this would be the case.  LIME, SHAP, and FFA measure *different things* with *different semantics* and *different properties*.  I'd like to understand why FFAs (which measure the chance that a certain feature occurs in the set of AXp's) are taken to be - essentially - *the* gold standard.  I could not find solid motivation for this in the text.  The authors write that FFAs are "[...] the percentage of (formal abductive) explanations that make use of a particular feature i", which I do agree with.  What I miss is a link between this and believing that FFAs are "better" in all respects than LIME or SHAP (two approaches that, honestly, I am not a fan of).

Q2. The authors stress the issue of "out-of-distribution sampling".  They mention this is an issue for AXp's because it yields complex explanations, hindering human understanding.  They also mention it is an issue for other, less "formal" attribution techniques like LIME and SHAP, but they don't explain why it is a problem.  My take is whathever the impact is on LIME and SHAP, it is not related to explanation complexity or length.  Then the authors explain that FFAs deal with explanation complexity, which I do agree with as FFAs compress the full set of AXp's into a fixed-size relevance vector.  However, I do not see how FFAs deal with the *other*, unnamed problem affecting LIME and SHAP.  TL;DR: what's the issue with OOD sampling and LIME and SHAP, and why wouldn't the same issue affect AXp's and/or FFAs?

Q3. My understanding is that each AXp is strongly determined by feature interactions.  Taking the average over AXp's entirely throws away this information.  I think this is an important limitation - not only of FFAs, but rather of all feature attribution techniques, chiefly due to their simple vector format - that should be openly discussed in the paper.  I am confident the authors are keenly aware of this issue, and I'd appreciate if they made the readers aware of it too.  To turn this into a question: would you agree that FFAs are by their own nature less expressive than (the set of) AXp's?

I am more than willing to increase my scores once the authors address these questions.

**Limitations:**

The paper has an explicit limitations section which is quite well done.  I have outlined a couple of other possible limitations in my questions.

---

> ### Author Rebuttal · Authors · 2023-08-09
>
> Thank you for the positive comments. As the weaknesses you identified
> directly correlate with the questions asked, let us address them by
> answering the questions down below:
>
> ### Answers to Questions
>
> **Q1.** Please see one of the general comments on the use of FFA
> and the adequacy of its comparison to LIME and SHAP above.
>
> **Q2.** We would like to clarify that out-of-distribution sampling
> represents an issue for sampling-based explainers but not for formal
> explainers. Namely, out-of-distribution sampling appears if instance
> perturbation performed by an explainer ignores the actual data
> distribution and so the explainer creates erroneous instances and so,
> as a result, faulty conclusions. We do not elaborate on this issue in
> the paper because it is quite known in the XAI literature since at
> least [62].
>
> AXp's are not susceptible to OOD, because the computation of an AXp is
> done by means of formally reasoning about the logical representation
> of the model and so formally proving that a subset of features is an
> AXp, using equation (1) and considering the entire feature space. No
> sampling is used here. Essentially, we prove that there is *no single*
> counterexample instance in the feature space that would invalidate our
> AXp. One the one hand, this is a very strong guarantee that makes
> AXp's "bullet-proof". On the other hand, this also makes AXp's quite
> bulky as we need to account for all possible combinations of feature
> values (and hence keep a large number of features in). This issue is
> one of the motivations for FFA.
>
> **Q3.** We agree with you that feature attribution approaches in
> general lack the information on feature interactions that may be
> available in feature selection explanations. Therefore, we do believe
> numeric values of FFA may be practically less informative than the set
> of AXp's used to generate these FFA values. Having said that, we
> should note that in stark contrast to the existing feature attribution
> approaches, the FFA computation and approximation method does not
> throw the evidence away - the AXp's can be kept and provided to a user
> on demand if additional evidence or insights turn out to be necessary.
> This is another advantage of FFA over the existing feature attribution
> approaches, which will surely comment on in the final version of the
> paper. Thank you!
>
> ### On Experiments with Boosted Trees
>
> Formal explanation enumeration, which our approach makes use of,
> requires one to represent the model of interest in a suitable logical
> formalism and then formally reason about its behaviour by means of
> making a series of reasoning oracle calls (note that this is also
> mentioned in the section on limitations). While for some kinds of
> models formal explanation extraction is computationally trivial to do,
> e.g. for decision trees or monotone classifiers, the other extreme is
> represented by various kinds of neural networks where formal
> explanation extraction is computationally quite challenging due to the
> lack of effective reasoners that could tackle the entire feature space
> with sufficient ease. As a result, we opt for a compromise between the
> power and generalizability of a model on the one side and
> computational tractability of formal explanation extraction and
> enumeration on the other side provided by tree ensembles. Note that as
> better formal reasoning methods get created for more kinds of ML
> models, FFA becomes more practically computable.
>
> ### On MNIST-Related Data
>
> Also, although we do evaluate approximate FFA computation only on
> MNIST-related data, it is important to observe that exact FFA is shown
> to be within the reach of our methods for numerous tabular data widely
> studied in the XAI domain.

---

> > ### Comment · Reviewer_FoWS · 2023-08-19
> > **Regarding the authors' response to Q3**
> >
> > Also in SHAP, it's possible to retain the intermediate outcomes, which can provide insights into interactions. If I grasp correctly, taking an average might obscure interaction effects, which could be crucial information. Moreover, AXp alone might not inherently unveil these effects; instead, they might require additional investigation based on the collected AXp data (which again, as I said, holds for SHAP as well).
> > If your method claims to offer an advantage in identifying significant interactions, it would be beneficial to provide a more detailed explanation of how exactly it achieves this.

---

> > > ### Comment · Reviewer_gWVL · 2023-08-19
> > > **Reply to authors**
> > >
> > > Thank you for the detailed reply.  A couple of remarks:
> > >
> > > Q1. I am aware that AXps have many selling points - from both a formal and computational perspective.  I never doubted this.  What I do not like is - again - that they are treated as a gold standard. Doing so essentially implies that not better type of explanations exists or, more subtly, that what matters are in fact the formal and computational properties of explanations.  Human factors matter just as much.  Do people understand FFAs better than explanations output (and defined) by LIME, SHAP or other attribution methods?  We don't really know. So it's better to stray on the safe side.  To be clear, I'd strongly prefer if FFAs were treated as *one* sensible choice, rather than *the* only sensible choice, as the paper - to my eyes - seems to imply.  Nothing would be lost by doing this.  Regardless, I don't think is not enough reason to reject a paper - and in fact note that my score is positive.
> > >
> > > Q2. Thank you for the clarification.  I will **increase** my score accordingly.
> > >
> > > Q3. I agree, this is a good point.

---

> > > > ### Author Response · Authors · 2023-08-21
> > > > **Reply to Reviewer gWVL**
> > > >
> > > > Regarding your comment on Q1, we are certainly not suggesting that FFA
> > > > is **the only possible** feature importance measure to apply, nor do
> > > > we say that formal abductive and contrastive explanations are the only
> > > > feature selection approaches to consider *in practice*. We acknowledge
> > > > the fact that besides formal explanations (including probabilistic
> > > > ones), there is a large body of work on sampling-based explanations
> > > > offering another way of understanding how features interact with a
> > > > model's predictions. Among these the idea to approximate Shapley
> > > > values stands out. But Shapley values should not be conflated with
> > > > feature importance. More on this, in our view, the main upside of
> > > > model-agnostic approaches seems to be what makes them fall short.
> > > > Namely, they are completely disentangled from the model's internal
> > > > workings and, hence, do not even agree on feature relevancy with the
> > > > formal semantics of the classifier they purport to explain.
> > > >
> > > > Here we should also say that we are well-aware of the limitations of
> > > > formal explanation approaches, including FFA's, and we list them in
> > > > the paper. One of them is the need to formally reason about the
> > > > model's behavior and, as a result, limited scalability. As users of
> > > > the AI/ML technology, we will surely continue using sampling-based
> > > > approaches, e.g. in situations where AXp/CXp/FFA approaches are not
> > > > applicable (for instance, when they don't scale). But we believe we
> > > > should be knowledgeable about the pitfalls of sampling-based
> > > > explanations and be careful with the conclusions we can draw from
> > > > them.
> > > >
> > > > Having said that, we absolutely agree with you that formal correctness
> > > > of explanations is not the only factor to consider and the human
> > > > factor you mention is practically *at least* as important as
> > > > explanation correctness. But you hopefully also agree that a user
> > > > should be in a much better position judging whether an explanation is
> > > > sensible to them if being conscious whether or not they can trust it.

---

> > > ### Author Response · Authors · 2023-08-21
> > > **Regarding the reviewer's FoWS additional comment on Q3**
> > >
> > > Thank you for the additional comment. We would like to point out that given a set of AXp’s collected, there are multiple ways to use them. For example, we can compute exact or approximate FFA. If a user needs evidence for the FFA values reported, some (or all) AXp's may be given to the user. Also, we can determine positive or negative feature correlation based on AXp's. We can also use CXp’s computed in the enumeration process to provide additional insights on features that contribute to changing the prediction. Overall, there is certainly much more to explore here and we believe FFA is a starting point for understanding the relationship of features to the prediction, which at least agrees with feature relevancy.

---

### Official Review · Reviewer_aL7B · 2023-07-07

**Soundness:** 2 fair
**Presentation:** 2 fair
**Contribution:** 2 fair
**Rating:** 5
**Confidence:** 4

**Summary:**

This work proposes a new approach called formal feature attribution (FFA), inspired by successful FXAI methods, to compute feature attribution scores. FFA is defined as the proportion of explanations where a feature occurs. Experiments try to demonstrate the effectiveness of FFA compared to SHAP and LIME under several public datasets.

**Strengths:**

1. The authors proposed a new perspective on providing feature attributions. They did point out some limitations of existing popular methods.
2. The proposed FFA is straightforward and with clear motivations.


**Weaknesses:**

1. Important questions regarding the reasonableness of the proposed method are unanswered. For instance, the three advantage properties of FFA claimed in this paper are not well reasoned.
2. The experiments provided in this work are hard to support FFA is better than other existing XAI scoring methods.
3. Several definitions and annotations throughout the text lack proper illustration or clarification, which can hinder the reader's understanding. Without clear explanations, it becomes challenging to grasp the intended meanings and implications of these terms and annotations.


**Questions:**

1. The authors claim that FFA gains three nice properties. However, the descriptions of those properties are too vague to understand. The authors are highly encouraged to explain more details about those points. Some example concerns are illustrated as follows. The first advantage is that FFA has a strict and formal definition. However, the author fails to explain why the Shapley Value lacks a strict and formal definition, despite it being commonly regarded as the ground truth value in feature attributions. I am unable to discern the distinction between the Shapley Value and FFA in terms of a strict and formal definition, as well as why the Shapley Value cannot provide formal feature attributions. Additionally, claiming that percentage-wise feature attributions are more user-friendly without conducting user experiment studies or related experiments is unsubstantiated. Insufficient case studies on image data are inadequate to provide support.
2. There are a lot of advancements discussing the issues of out-of-distribution sampling. It would be better to see the comparison between these series of methods, making the proposed approach more convincing, as FFA is claimed to address the limitations of out-of-distribution sampling (see line 140 to 148).
3. LIME and SHAP are two classical baselines to provide a model explanation but obtain inferior performance on image datasets. It is highly encouraged the authors compare with some SOTA Shapley-based methods on image data, such as DeepSHAP[1], FastSHAP[2], or CoRTX[3].
4. As for the experimental results in Figure 3, it is very limited for authors to use only one example to reveal that FFA is better than LIME and SHAP. Furthermore, Table 1 only demonstrates LIME and SHAP fail to get close enough to FFA, which is expected as neither of them is designed to approximate FFA. However, the conclusion in Table 1 does not mean that FFA is a better indicator than LIME, SHAP, or Shapley Values in providing a formal explanation. I did not see any direct evidence to support this point.

[1] Delivering Trustworthy AI through Formal XAI. Marques-Silva et al. AAAI 2022.
[2] A unified approach to interpreting model predictions. Lundberg et al. NeurIPS 2017.
[3] FastSHAP: Real-Time Shapley Value Estimation. Neil et al. ICLR 2022.
[4] CoRTX: Contrastive Framework for Real-time Explanation. Chuang et al. ICLR 2023.


**Limitations:**

1. The proposed FFA only focuses on the classification task, which makes it limited to be applied in several other common tasks, such as the regression task.
2. The paper lacks clear illustrations of experiment settings and does not compare with relevant advancements, leading to unconvincing results for verifying the proposed hypothesis. More detailed descriptions of the experiment settings and comparative analyses with related advancements are necessary to strengthen the study's validity.
3. Several annotations and definitions are not clearly illustrated. For example, the “right arrow” in Equation 1 and the definition of “formal explanation” is not well-illustrated in this work. This makes the work hard to follow.

---

> ### Author Rebuttal · Authors · 2023-08-09
>
> Thank you for the comments. We try to address them below, which will
> hopefully convince you that our work has merits justifying acceptance.
>
> ### Answers to Questions
>
> **Q1.** We would like to clarify that by no means we claim that
> Shapley values have no formal definition. On the contrary, as Shapley
> values themselves originate from the work on cooperative games, they
> are formally well-defined in the original context. Having said that,
> we build on the evidence provided in recent work [21] that *(even
> exact)* Shapley values may assign non-zero feature attribution to
> features unrelated to the prediction and, the other way around, may
> assign zero attributions to features that play a role in the
> prediction process. We are not saying that we should stop using
> approximate Shapley values but the issues revealed in [21] are quite
> concerning and they motivate plausible alternatives to Shapley values,
> which FFA is an example of.
>
> **Q2.** Note that FFA is not claimed to target specifically the
> problem of out-of-distribution sampling. We mention OOD sampling as an
> *example issue* that most of the perturbation-based approaches are
> susceptible to and that *none* of the formal approaches exhibits.
>
> **Q3.** LIME and SHAP are examples of extremely successful explainers
> widely used in practice and *extensively cited* in the XAI literature.
> To our best knowledge, DeepSHAP does not support tree ensemble models
> and so we cannot compare against it. As for FastSHAP [a], we ran
> additional experiments with it, KernelSHAP [b] and its acceleration
> (denoted as KernalSHAP-S in the attached PDF) that uses paired
> sampling [c], and the results can be found in the attached PDF.
> Observe that these explainers perform similarly to LIME and SHAP,
> which is not surprising as they approximate exact Shapley values,
> which are known to be not that closely related to feature importance
> [21].
>
> We were unable to complete experiments using CoRTX in time, but we do
> not expect the results to be different since it again computes
> approximate Shapley values, not explanation attributions. Also, note
> that CoRTX was published in May 2023, i.e. around the time of NeurIPS
> submission.
>
> **Q4.** Please see our general rebuttal above on as to why we believe
> FFA is a good metric for feature attribution while prior work [21]
> revealed the issues pertaining to Shapley values in the explainability
> context (also see the answer to Q1).
>
> ### On Inconvincing Experiments
>
> Note that we use multiple datasets in our experiments demonstrating
> how much LIME and SHAP disagree with FFA. Figure 3 only serves as a
> single example taken for the Compas dataset, which is widely used in
> the XAI literature, while Table 1 provides average information across
> hundreds of instances explained for a number of datasets (including
> Compas). We can easily show many more examples but the page limit
> prevents us from doing so.
>
> ### On Classification Focus
>
> We would like to note that all the modern formal XAI (FXAI)
> approaches, without exception, aim at tackling classification
> explainability. Hence, this should not be treated as a limitation of
> our work - this is simply beyond the scope of FXAI in general (and our
> paper too) and it requires a large body of currently missing work in
> the area. Nevertheless, if we want to explain why a regression value
> sits within given lower and upper bounds then we can apply the same
> technology, with no changes, assuming we can logically represent the
> regressor, since the question has been converted to a true/false
> question.
>
> ### On Missing Details and Unverified Hypothesis
>
> We would like to kindly note that the experimental setup is described
> in the paper and the results are discussed within the limits of the
> page limit. Moreover, the entire experiment can be reproduced using
> the source code, the benchmarks and all the scripts provided in the
> supplementary material, which was submitted with the paper.
>
> ### On Presentation Issues
>
> We are happy to improve the presentation further if it helps a reader
> understand the ideas. The "right arrow" is the implication operator
> representing logical entailment widely used in mathematics, computer
> science, and other sciences. Namely, whatever is written to the left
> of the arrow logically entails whatever is written to the right of the
> arrow. Note that formula (1) is augmented with a short text
> description that summarizes the meaning of the formula.
>
> - [a] FastSHAP: Real-Time Shapley Value Estimation. Neil et al. ICLR 2022.
> - [b] A unified approach to interpreting model predictions. Lundberg et al. NeurIPS 2017.
> - [c] Improving KernelSHAP: Practical Shapley value estimation using linear regression.
> Covert, I. and Lee, S.-I. PMLR 2021.

---

> > ### Comment · Reviewer_aL7B · 2023-08-13
> > **Official Comment by Reviewer aL7B**
> >
> > Thank you for the hard work and effort you put in a very short time. It is very impressive. I do appreciate the clarification and experimental results provided by the authors. I now better understand the contributions of this work, and most of my concerns are addressed.
> >
> > The reason that I think FFA is trying to solve the OOD problem comes from Line 145, where the authors mention the goal of the paper is to solve the limitations mentioned in the previous paragraphs. And the previous paragraphs discuss the limitation of the OOD problem. This is confusing for readers to realize the contribution and motivation of FFA. Furthermore, FFA only applies to the classification task, which may limit its usage in more real-world applications. I think this is still a limitation in this paper, but a worthwhile future direction to solve. As a pioneering paper to discuss FFA, I will not consider this point as a drawback during the reviewing session.
> >
> > I have three more follow-up questions after rereading the paper again. I am willing to recommend acceptance if the follow-up questions are solved.
> >
> > **(1)** FFA is only tested on some naive and small datasets in classification tasks, such as the 28x28 size of MNIST and 28x28 PneumoniaMNIST. The size of the image data here is far away from the size in the real-world dataset. I would like to know why the authors chose not to evaluate under other image datasets, such as CIFAR-10 and ImageNet, with larger images. Maybe the high computational complexity of FFA prevents you from doing so? or are there any other reasons?
> >
> > **(2)** The experimental results presented in this study are assessed using the exact FFA value (please correct me if my understanding is inaccurate). Upon reviewing the outcomes of KernelSHAP, FastSHAP, and KernelSHAP-S (both Shapley-based methods), I believe that KernelSHAP and FastSHAP are expected to demonstrate inferior performance, as their primary focus is on estimating Shapley values rather than specifically targeting FFA estimation. On the other hand, ${FFA}_{numbers}$ are designed to estimate FFA, which is expectedly to have better performance with lower estimation error. I agree with Reviewer FoWS, and it would be better if this work could include the axiomatic analysis of FFA, as the experimental results are hard to convince. I would like to ask the authors to provide more insights or observations regarding this question.
> >
> > **(3)** What is the difference between SHAP and KernelSHAP? To the best of my knowledge, these two are the same baseline, so why do they obtain different performances?

---

> > > ### Author Response · Authors · 2023-08-17
> > > **On Three Additional Questions from Reviewer aL7B**
> > >
> > > We are happy the clarification helped! We will rephrase the sentence
> > > on OOD (line 145) in the final version of the paper so that it does
> > > not confuse our readers. Thank you! As for your 3 additional
> > > questions, let us reply below:
> > >
> > > #### **1. On Using MNIST**
> > >
> > > It is true that we tested FFA on relatively small datasets but, to
> > > the best of our knowledge, those are quite widely used in the XAI
> > > literature (this applies to both tabular and image data we used).
> > > Thank you for pointing out these two image datasets.
> > >
> > > To be honest with you, we overlooked the possibility to test FFA
> > > approximation on CIFAR-10 while we could do it - nothing prevents us
> > > from testing it except for time. While we do not anticipate the same
> > > level of efficiency as with MNIST due to the (slightly) larger size of
> > > CIFAR-10, we acknowledge the potential significance of including them.
> > > We will consider larger image datasets, such as CIFAR-10, in future
> > > work.
> > >
> > > Having said that, ImageNet should definitely be out of reach for our
> > > method. This is because FFA approximation requires enumerating
> > > abductive explanations, which in turn requires efficient formal
> > > reasoning about a model's behavior. Given the size of ImageNet, the
> > > models to use here may be too large to handle by formal reasoning. It
> > > would be interesting to see what models could be effectively trained
> > > on ImageNet and what accuracy they might have, to check if any of them
> > > were within the reach of modern formal methods.
> > >
> > > #### **2. On SHAP's Behavior vs. FFA Approximation**
> > >
> > > Your understanding is correct that a large portion of the original
> > > experimental results and all the additional results are obtained for
> > > the cases, where we can efficiently compute *exact FFA*. Although we
> > > surely agree with you and the other reviewers that SHAP-like
> > > explainers *are not designed* to approximate FFA, it seems valid to
> > > compare the values they report with those of FFA. Our findings show
> > > that feature attributions calculated by LIME and multiple versions of
> > > SHAP are far away from FFA. While we well understand the meaning of
> > > FFA, a very simple metric, we believe these observations should
> > > motivate the community to think over what these explainers actually
> > > compute in practice and how to treat them from different perspectives,
> > > e.g. from the view of formal reasoning.
> > >
> > > As for an axiomatic analysis, we agree this would be interesting to
> > > investigate. Please also see our response to Reviewer Cksz, where some
> > > initial observations on this are provided. Having said that, we
> > > believe our work can be seen as a starting point in this direction,
> > > motivating further effort in the near future.
> > >
> > > #### **3. On Difference Between SHAP, KernelSHAP, and KernelSHAP-S**
> > >
> > > - "KernelSHAP" used in the additional experiments provided upon your
> > > request denotes one of the original versions of SHAP presented in
> > > [40].
> > >
> > > - The version of "SHAP" used in our paper is an *improved* version
> > > published by the same authors in [d], which refers to it as TreeSHAP.
> > > We used it because it is designed to better handle tree-based models
> > > including random forests and tree ensembles like XGBoost and LightGBM,
> > > and other. To our best knowledge, compared to KernelSHAP, it directly
> > > uses the structure of the tree models when sampling and, therefore, it
> > > is claimed to be advantageous to KernelSHAP (recall that KernelSHAP is
> > > model-agnostic) in two aspects:
> > >
> > >     - In TreeSHAP, instead of iterating through all possible feature
> > >   combinations (or a subset thereof), each combination is processed
> > >   concurrently within the tree. It uses a more complex algorithm to
> > >   monitor the results of each combination and the overall complexity
> > >   is reduced. Therefore, TreeSHAP eliminates all the sampling-based
> > >   estimation variance and it is not required to use a background
> > >   dataset or select a subset of feature combinations.
> > >
> > >     - The Shapley values computed by TreeSHAP are not skewed due to
> > >   feature dependencies, as these dependencies are contained within the
> > >   tree structure.
> > >
> > > - "KernelSHAP-S" used in the additional experiments is an accelerated
> > > version of "KernelSHAP", which applies paired sampling [c] (the reference
> > > can be found in our earlier reply).
> > >
> > > [d] From local explanations to global understanding with explainable
> > > AI for trees. Lundberg et al. Nature machine intelligence, 2020.
> > >
> > > As a result, the approximations of Shapley values reported by these
> > > three versions of SHAP and different and their performance also
> > > differs.

---

> > > > ### Comment · Reviewer_aL7B · 2023-08-17
> > > > **Official Comment by Reviewer aL7B**
> > > >
> > > > Thank you for the response. I think most of my concerns are addressed. I will raise my score accordingly. SHAP and KernelSHAP are typically represented as the same baseline, which readers can easily confuse. Please ensure to include all the information in the revised version, including the difference between SHAP and KernelSHAP.

---

> > > > > ### Author Response · Authors · 2023-08-17
> > > > >
> > > > > Thank you! We will surely add all the promised clarifications in the final version of the paper.

---

### Author Rebuttal · Authors · 2023-08-09

We thank the reviewers for the thorough and helpful comments.

### Why FFA?

Several reviewers raise concerns regarding the use of FFA as a "gold
standard" in feature attribution and also regarding the validity of
its comparison with other feature attribution measures. Hence, we
would like to give a few general comments on this.

The key insight that our definition of FFA builds on is that formal
abductive explanations are (*provably* guaranteed) reasons for the
predictions made by a given model. Essentially, if a subset of
features is claimed to be an AXp for a prediction made by our model
then we can be certain that assigning these features to the values
dictated by the instance will *necessarily* lead to the same
prediction. Thanks to the subset-minimality of AXp's, we can also be
certain that nothing can be removed from an AXp, i.e. none of its
proper subsets is an AXp. Note that this is not estimated
statistically but it is rather *proved formally* based on the logical
representation of the model of interest, which makes the concept very
strong.

Given this, complete enumeration of all possible AXp's for a model's
prediction allows us to explore all the subsets of feature-values that
logically entail this concrete prediction. We emphasize that upon
completion of AXp enumeration, *no other logical reasons exist* for
this concrete model's prediction.

This enables us to investigate feature relevancy, i.e. a feature is
deemed relevant for a given prediction if it belongs to at least one
AXp for that prediction; and vice versa, if it does not then it is
irrelevant for the prediction. This also enables us to estimate how
important a feature is, i.e. how frequently it appears in explanations
across *all* the AXp's for this prediction.

In this regard, let's consider two types of attribution errors: TYPE1
where an attribution method says the feature is relevant while it is
not, and TYPE2 where the method says that the feature is irrelevant
while it is relevant. Importantly, our method never makes TYPE1
errors, even when approximating FFA; this is not the case for the
competitors.

Consider our example in the paper (see Examples 1-3 and Figures 1-2).
Given an instance to explain: *{"Education"="Bachelors",
"Status"="Separated", "Occupation"="Sales",
"Relationship"="Not-in-family", "Sex"="Male", "Hours/w"<=40}*, we
enumerate all AXp's - there are two of them: {"Education", "Hours/w"}
and {"Education", "Status"}. As long as these features are set to the
values of the instance, we are sure the model will predict "< 50k" no
matter what other features are assigned to. Based on this, we can say
that feature "Education" is the most important as it appears in all
AXp's while "Hours/w" and "Status" have importance of 0.5 as each of
them appears in a half of the AXp's. All the other features are
irrelevant for this prediction.

To illustrate this further, consider the following decision set of 12
irreducible rules, which is **equivalent** to the tree ensemble shown
in the paper:

```
01: IF 'Status == Never-Married' AND '40 < Hours/w <= 45' THEN '< $50k'
02: IF 'Status == Never-Married' AND 'Relationship != Not-in-family' THEN '< $50k'
03: IF 'Education != Doctorate' AND 'Status != Married' THEN '< $50k'
04: IF 'Education != Doctorate' AND 'Hours/w <= 40' THEN '< $50k'
05: IF 'Education != Doctorate' AND 'Relationship == Own-child' THEN '< $50k'
06: IF 'Education == Dropout' THEN '< $50k'
07: IF 'Status != Married' AND 'Relationship != Not-in-family' AND 'Hours/w <= 45' THEN '< $50k'
08: IF 'Education == Doctorate' AND 'Relationship == Not-in-family' AND 'NOT 40 < Hours/w <= 45' THEN '>= $50k'
09: IF 'Education == Doctorate' AND 'Status != Never-Married' AND 'Relationship == Not-in-family' THEN '>= $50k'
10: IF 'Education != Dropout' AND 'Status == Married' AND 'Relationship != Own-child' AND 'Hours/w > 40' THEN '>= $50k'
11: IF 'Education == Doctorate' AND 'Status == Married' THEN '>= $50k'
12: IF 'Education == Doctorate' AND 'Status != Never-Married' AND 'Hours/w > 45' THEN '>= $50k'
```

We emphasize that this DS replicates the behavior of the original BT
model in the *entire* feature space. The only two rules applicable to
the considered instance are `03` and `04`. In fact, they determine the
two AXp's shown above, which we believe provides very clear grounds
for defining FFA.

Observe that comparing FFA with the explanations of LIME and SHAP is
adequate because they also claim to measure feature importance for a
given model's prediction, based on statistical observations during
extensive sampling in the instance's vicinity. Clearly, statistical
correlation does not represent a *causal premise* for the prediction
and in practice it may often lead to attributions having little to do
with reality. This is confirmed by our example (and experimental
results *in general*): observe how both LIME and SHAP claim non-zero
importance of feature "Relationship" even though it does not appear in
any AXp, a TYPE1 error. Importantly, this feature has nothing to do
with the prediction for our concrete instance, according to the
equivalent (and simple to understand) DS model, which is at least
puzzling. Similarly, both LIME and SHAP fail to appreciate the
importance of feature "Education" (TYPE2 error).

### Additional Experimental Results

As Reviewer aL7B requested, we ran additional experiments with
FastSHAP, KernelSHAP and KernelSHAP-S on the benchmarks used in the
paper. The use of SHAP variants results in feature attributions
roughly similar to those of LIME and SHAP. It is not surprising given
that the nature of these explainers is similar to that of SHAP, i.e.
they try to approximate Shapley values, which are not that related to
feature importance [21]. Apart from FastSHAP, each of these variants
makes TYPE1 error on the the above example, ascribing non-zero
importance to "Relationship". FastSHAP makes the same TYPE2 error as
SHAP.

---

### Decision · Program_Chairs · 2023-09-21

**Decision:**

Reject

**Comment:**

This paper generated discussion both between the reviewers and the authors and between the reviewers themselves. Overall, all agree that this work has merits. However, most reviewer thought that the novel ideas should be further developed before publication. For example, it claims to be formal but essentially the theory just reiterates the definitions. At the same time, some reviewers were very appreciative of the ideas presented here, even if the evaluation is limited, however the decision was that it does not cross the bar for a publication in NeurIPS.